# Recognition of Arabic Air-Written Letters: Machine Learning, Convolutional Neural Networks, and Optical Character Recognition (OCR) Techniques

**DOI:** 10.3390/s23239475

**Published:** 2023-11-28

**Authors:** Khalid M. O. Nahar, Izzat Alsmadi, Rabia Emhamed Al Mamlook, Ahmad Nasayreh, Hasan Gharaibeh, Ali Saeed Almuflih, Fahad Alasim

**Affiliations:** 1Computer Science Department, Faculty of Information Technology and Computer Sciences, Yarmouk University, Irbid 21163, Jordan; nasayrahahmad@gmail.com (A.N.); hasangharaibeh87@gmail.com (H.G.); 2Department of Computing and Cyber Security, Texas A&M University-San Antonio, San Antonio, TX 78224, USA; izzat.alsmadi@tamusa.edu; 3Department of Business Administration, Trine University, Angola, IN 49008, USA; 4Department of Mechanical and Industrial Engineering, University of Zawia, Tripoli 16418, Libya; 5Department of Industrial Engineering, College of Engineering, King Khalid University, Abha 62529, Saudi Arabia; asalmuflih@kku.edu.sa; 6Department of Industrial Engineering, College of Engineering, King Saud University, Riyadh 11495, Saudi Arabia; falasim@ksu.edu.sa

**Keywords:** Arabic air-writing recognition, machine learning, OCR, recognition, deep learning

## Abstract

Air writing is one of the essential fields that the world is turning to, which can benefit from the world of the metaverse, as well as the ease of communication between humans and machines. The research literature on air writing and its applications shows significant work in English and Chinese, while little research is conducted in other languages, such as Arabic. To fill this gap, we propose a hybrid model that combines feature extraction with deep learning models and then uses machine learning (ML) and optical character recognition (OCR) methods and applies grid and random search optimization algorithms to obtain the best model parameters and outcomes. Several machine learning methods (e.g., neural networks (NNs), random forest (RF), K-nearest neighbours (KNN), and support vector machine (SVM)) are applied to deep features extracted from deep convolutional neural networks (CNNs), such as VGG16, VGG19, and SqueezeNet. Our study uses the AHAWP dataset, which consists of diverse writing styles and hand sign variations, to train and evaluate the models. Prepossessing schemes are applied to improve data quality by reducing bias. Furthermore, OCR character (OCR) methods are integrated into our model to isolate individual letters from continuous air-written gestures and improve recognition results. The results of this study showed that the proposed model achieved the best accuracy of 88.8% using NN with VGG16.

## 1. Introduction

Advances in information technology have reshaped how humans interact with machines and programs and communicate within their environment and language. Writing in the air has emerged as a mode of communication between humans and their intelligent applications and devices, aligning with the flexibility of human mobility and surroundings [1]. This air-writing modality finds utility across diverse fields, including human–robot communication, children’s education, aiding individuals with sensory challenges, and even in the realm of the metaverse [2]. Recognizing air-written Arabic letters introduces a tapestry of intricate challenges necessitating inventive solutions. Diverging from traditional written scripts, the fluid nature of air-written gestures presents layers of complexity that warrant specialized approaches. The dynamic essence of air-written letters transcends the confined spatial boundaries of conventional paper or screens, engendering a rich spectrum of letter formations and trajectories. Unravelling and deciphering these ever-evolving spatial–temporal patterns emerge as a formidable hurdle. Further intricacy arises from individuals’ various writing styles and hand sign variations. This myriad of expressions compounds the complexities tied to precise interpretation and recognition of air-written Arabic letters. Furthermore, restricting distinct individual letters within the continuum of air-written gestures mandates meticulous segmentation techniques for unwavering recognition accuracy. Amplifying the challenge is the absence of predetermined reference points or demarcated start and end strokes in air writing, necessitating ingenious algorithms capable of impeccably detecting letters despite the lack of conventional structural cues. While the recognition of air-written English letters has undergone thorough exploration using a range of machine learning techniques, including LSTM [3], 2D-CNN, Faster RCNN [4], and 3D Reset [5], a notable gap exists in the realm of recognizing air-written Arabic letters. Despite the extensive research on English counterparts, studies have yet to venture into deciphering air-written Arabic letters. In this complex landscape, our approach orchestrates a holistic methodology harmonizing the strengths of machine learning (ML) methods, deep convolutional neural networks (CNNs) [6], and the finesse of optical character recognition (OCR) techniques [7]. OCR, a process that transforms images of handwritten or typewritten text into machine-readable text, forms a key component of our strategy. This is aligned with broader endeavours to process handwritten text from electronic devices, such as paper forms, invoices, and legal documents. However, grappling with a blend of machine-generated and human-written text poses distinct challenges, particularly in the public or governmental sectors. Effective handling and storage of such diverse inputs remain an ongoing concern, wherein OCR plays a pivotal role by converting text images into machine-readable text data. The driving force behind this research is rooted in the inherent capability of air-writing recognition to bridge the divide between genuine human gestures and digital interfaces. As intelligent systems weave more intricately into our daily lives and the demand for seamless interaction burgeons, the capacity to decode and comprehend air-written letters assumes a pivotal significance. Moreover, the allure of Arabic script, epitomized by its intricate elegance, confers an additional layer of attraction to this quest, urging us to embark on a profound exploration within this uncharted territory. These research efforts aim to make a meaningful contribution by teaching Arabic to non-native speakers and providing interactive training for young learners to master the art of drawing Arabic letters. The intention is to foster an engaging experience with technology that stimulates curiosity and ignites creativity. Moreover, this technology has the potential to assist individuals with speech challenges. Many individuals who struggle with speech can still write, and this application could be harnessed to track hand movements and convert them into synthesized human speech. It opens new avenues of communication and expression for those facing obstacles in traditional verbal interaction. The main objective of this research is to construct an intricate model that seamlessly intertwines machine learning and optical character recognition techniques. This model is engineered to accurately identify individual Arabic characters and entire words formed in the air. By achieving this ambitious target, this research can significantly enhance Arabic language education and foster more inclusive and efficient modes of communication. The rest of this paper is organized as follows: Section 2 reviews the existing literature on air-written Arabic letter recognition. Section 3 presents a methodology for developing and evaluating the recognition model. Section 4 summarizes the results and analysis of the recognition model. Section 5 concludes this paper by discussing the implications of the research findings.

## 2. Related Work

Researchers have shown considerable interest in air-writing recognition, extending beyond just numerical digits and symbols to encompass various languages. In this section, we delve into prior investigations that have delved into air-writing recognition, explicitly focusing on numbers, symbols, and linguistic diversity.

### 2.1. Air Writing with Numbers and Symbols

A particularly noteworthy investigation, Ref. [2] utilized radar-based methodologies to track the trajectories of hand motions while individuals engaged in the act of air-writing numerical digits. A multi-stream convolutional neural network (MS-CNN), coupled with continuous wave radar frequency, was enlisted to distinguish numbers ranging from zero to nine. Impressively, this model achieved an accuracy of 95% when tested with numerals air-written by 12 volunteers, shedding light on the potential of radar-based approaches in the realm of air-writing recognition. Beyond numbers, air writing encompasses hand-drawn symbols, holding significance across various applications such as encrypted codes and authentication systems. Another study [8] introduced an algorithm to extract the trajectories traced during air writing, accompanied by the development of deep CNNs, encompassing both 1D-CNN and 2D-CNN architectures for deciphering hand-drawn symbols. Diligent parameter optimization contributed to achieving a high discrimination rate of 99% using their CNN models. The dataset used in their investigation covered numeric symbols (zero to nine) drawn in both clockwise and counterclockwise directions, along with a set of 16 directional symbols. The models underwent training and testing on well-segmented datasets, utilizing K-fold cross-validation to determine the optimal K value. A recognition rate of five yielded the most favourable results, underscoring the effectiveness of deep CNNs in symbol recognition. Given the increasing ubiquity of smart devices, gesture-based communication has gained prominence in interactions with these devices. Addressing this trend, another study [3] directed their attention towards enhancing hand gesture communication with air writing. They devised a system for recognizing air-typed gestures, combining 3D trajectories with a fusion of long-term memory (LSTM) and convolutional neural networks (CNNs). Preprocessing techniques such as normalization and root point translation were applied to the trajectory data. The evaluation was conducted using a dataset of 2100 numerals gathered by the authors, resulting in an impressive accuracy rate of 99.32%. While these studies represent significant progress in air-writing recognition concerning numbers and symbols, exploring air-writing recognition within the Arabic language remains relatively uncharted. This study takes on the challenge of bridging this gap by developing a model carefully tailored to distinguish air-written Arabic letters. Using a synergy of machine learning techniques and optical character recognition, our objective is to establish a benchmark of precise and robust recognition performance within the domain of Arabic air writing.

### 2.2. Exploring Air-Written Letters

Deciphering air-written letters presents unique challenges, particularly when considering gesture recognition and handwriting analysis. Various research endeavours have delved into methods to identify and deconstruct handwritten or typed letters in the air. These efforts focus on accurately segmenting words and recognizing individual characters within air writing. A notable attempt by [9] involved using hashing techniques and a CNN model to fragment letters, resulting in an impressive 92.91%. Similarly, in a study by [1], a 2D-CNN model excelled in identifying letters and numbers written in the air, outperforming alternative methods. This highlights the efficacy of deep learning techniques in recognizing air-written letters and numbers. Another research initiative led by [4] compiled a dataset featuring hand movement videos in diverse settings and devised a recognition system for air-written letters. To train their dataset, they harnessed pre-trained models like Single Shot Detector (SSD) and Faster RCNN, achieving remarkable accuracy of up to 99%. Their work aimed to enhance the diversity and performance of prior datasets. Within the domain of optical character recognition (OCR), Ref. [10] implemented a CNN and RNN-based OCR system for recognizing diacritics and Ottoman font in Arabic script. This effort yielded remarkable outcomes: a validation accuracy of 98%, a word recognition rate (WRR) of 95%, and a character recognition rate (CRR) of 99% on the test dataset. Furthermore, endeavours have been directed towards recognizing handwritten Arabic letters in specific contexts. One study introduced ref. [11] a segmentation algorithm for handwritten Urdu script lines, achieving an accuracy of 96.7% for handwritten text and 98.3% for printed text. Ref. [12] proposed a technique grounded in the Faster RCNN framework for detecting and segmenting hand postures during air typing initiation, leading to a 96.11% character recognition accuracy. Additional studies have tackled the recognition of Arabic letters in varied scenarios. Ref. [13] presented an optical system for character recognition that utilized the learning vector algorithm and classification techniques to discern well-written and poorly written Arabic letters, with recognition rates of 86.95% and 54.55%, respectively. Ref. [14] constructed a comprehensive OCR system relying on CNN and SVM, achieving a recognition rate of 99.25% for Arabic letters and numbers. Ref. [15] compared diverse classifiers, including SVM, kNN, Naïve Bayes, ANN, and ELM, to classify an array of gestures, attaining the highest accuracy of 96.95% with SVM. The paper [16] proposes a system for recognizing characters written with fingers using time-of-flight (ToF) distance sensors, which works to capture the distance values between the writing finger and the sensors. The approach was tested on 26 English letters collected from 21 people, where the LSTM algorithm achieved an accuracy of 98.31%.

Nevertheless, despite these strides, certain gaps persist in the literature concerning air-written Arabic letter recognition. A prevalent issue arises from researchers using their distinct datasets, which complicates model evaluation due to the need for comparable benchmarks. Additionally, numerous studies have evaluated models on limited sample sizes, potentially needing to reflect real-world conditions more adequately. Further research is warranted in exploring advanced deep learning algorithms, such as convolutional neural networks (CNNs) and recurrent neural networks (RNNs), to enhance accuracy and resilience in recognizing air-written Arabic letters. While traditional machine learning techniques like SVM, Naïve Bayes, and ANN have been explored, there is room for further investigation. There is a lack of work on the Arabic language due to the difficulty of working on it and prepossessing. In addition, there needs to be more use of optimization algorithms to obtain optimal results. Moreover, air writing needs more work in general despite the positives that it can provide in various fields. This research aims to practically contribute to the advancement of gesture-based interaction, Arabic language education, and communication systems for individuals with disabilities. This pursuit aims to devise a scheme that facilitates effective and precise communication between humans and computer systems using air-written Arabic letters. By achieving these goals, this research seeks to enhance the identification of Arabic air-written letters by amalgamating machine learning, deep CNNs, and OCR approaches. Ultimately, developing a reliable and accurate system capable of detecting and comprehending air-written Arabic characters across diverse applications can enhance human–machine interaction and broaden access to learning and interacting in Arabic.

## 3. Arabic Air Writing to Image Conversion and Recognition: Methodology

This study presents a model to recognize writing in the air by identifying the hand’s boundaries and turning the writing into an image. In our experiments, we used the AHAWP dataset [17]. Two models were created; the first relies on writing a single letter, and the second relies on writing a word in its whole. We tested extracting the crucial features from the photos for a single word using deep learning methods and training models like VGG16, VGG19, and SqueezeNet. Then, normalization was used to enhance performance, lessen sensitivity, and increase model stability. Support vector machine (SVM), random forest (RF), K-nearest neighbour (KNN), and neural networks (NN) were then used as classification techniques for handwritten messages, as well as I2OCR for verification. The second model is based on predicting the completed handwritten word with the aid of I2OCR. Figure 1 displays the architecture of the proposed model, where the first step is to prepare the dataset and apply the prepossessing methods to it. After that, the three pre-trained models were used to extract the features from the images, then four machine learning were used for the classification of the eighteen Arabic letters.

### 3.1. AHAWP Dataset

In this research, we utilized the AHAWP dataset [17], which consists of letters, words, and paragraphs. The dataset was collected from 82 individuals, including 9000 images of alphabets. These images display letters, in positions within words, such as at the beginning, middle, or end. The dataset encompasses a total of 18 letters. To conduct our experiments, we split the dataset into a training set comprising 80% of the data and a testing set including the remaining 20%. This division allowed us to train our models on a portion of the data while also preserving a set for evaluating their performance. Figure 2 visually represents a sample from the AHAWP dataset.

### 3.2. Data Preprocessing

Preprocessing enhances dataset quality by eliminating irrelevant data and preparing it for model development. Our study focused on the processing steps for recognizing Arabic air-written letters. These steps play a role in preparing the dataset for analysis and classification. Specifically, we carried out the following processing steps.

#### 3.2.1. Image Prepossessing

The stage of processing images is considered one of the necessary stages before entering them into deep learning models to extract features from those images. The images were initially refined to clarify the letters and their edges, which helps extract important features. Noise was removed from the images. To ensure consistency between all samples, we changed the size of the serif letters to a size such as (224, 224). This resizing process ensured that all character images had dimensions and facilitated feature extraction and analysis. We used libraries like OpenCV or PIL to perform the processing on the images. There is no doubt that the augmentation process is considered one of the preferred processes in image processing, especially in increasing the number of data points. Still, in our study, we avoided using it because of changing a letter’s structure if it is flipped or skewed.

#### 3.2.2. Feature Extraction

Moving forward in our study, we drew pertinent details from the meticulously prepared character images. To accomplish this, we used a widely recognized method known as discrete cosine transformation (DCT), which has a solid feature extraction reputation. The DCT takes hold of the image by translating it into an ensemble of frequency coefficients that manage to encapsulate the crucial patterns and alterations inherent in the characters. This feature extraction aimed to capture discerning cues that could serve as valuable guides in the forthcoming classification endeavour.

#### 3.2.3. Dimensionality Reduction

With the features in hand, our next step involved reducing the complexity of the feature space using dimensionality reduction. A reliable method in this field is principal component analysis (PCA) [18], a technique that has been used to reduce the computational burden and avoid less informative features. In our exploration, one of the important factors in PCA is choosing the number of PCs [19]. We simplified the number of features from 4095 for the VGG16 and VGG19 models and 1000 for the SqueezeNet model to 99 features based on the variance value. That is, the lower the variance value, the more we preserved the original value of the features. This reduction in the number of features creates a means for simplified analysis and efficient classification of our data.

#### 3.2.4. Data Normalization

In order to put the data on the same playing field and set the stage for optimal machine learning performance, we delved into the realm of data normalization. Our strategy focused on refining our reduced feature vectors, ensuring they marched in harmony within a shared range, perhaps straddling the confines of −1 and 1. This normalization choreographed a balancing act, achieved using techniques like the tried-and-true min–max scaling or trustworthy standardization. By taking this stride, we bid farewell to any biases stemming from dissimilar feature scales, ushering in a level ground for judicious comparisons as we transition into the eagerly awaited classification phase. We adopted an 80% training to 20% testing data split approach to ensure better-trained models.

### 3.3. Building Air Writing Components

The proposed method consists of several stages for building the air writing components, namely: (1) the development of air writing tools, (2) the implementation of i2OCR, (3) feature extraction, and (4) classification.

#### Air Writing Tools

The proposed approach expects the user to start by drawing on a canvas using hand gestures, which can be detected with a webcam and Google’s Mediapipe library for body key point detection. It initializes four arrays (bpoints, gpoints, rpoints, and ypoints) to store points of different colours (e.g., blue, green, red, and yellow). These arrays are implemented as deques with a fixed maximum length of 1024. It also initializes four variables (blue index, green index, red index, and yellow index) to keep track of the current index in each array. The kernel (a morphological image processing operation for dilation and an array of colour tuples) is then defined. A variable colour index is initialized to store the current colour index. Canvas is then constructed by creating a blank image and displaying it in a paint window, as shown in Figure 3.

The next step is to initialize the media pipe hands, draw modules, and set up the webcam for capturing frames. In the main loop of the script, each frame is read from the webcam, flipped vertically, and converted to RGB colour space. UI elements are then drawn on the frame, including rectangles and text labels for the different colours and a ‘CLEAR’ button, as shown in Figure 3. The RGB frame is then passed to the media pipe hands module to detect hand gestures. If a hand is detected, the code draws a bounding box around the hand and obtains the hand’s key points (joints). The system then needs to check if the user has selected a colour by clicking one of the colour buttons. If a colour is selected, the code obtains the coordinates of the hand’s palm and appends them to the appropriate colour array. It also dilates the points in the colour array to make them thicker on the canvas after starting to write, and on the (B) side, signs to stop writing. Figure 4 shows the result of writing. After completing the writing, the final result appears on a white screen, as shown in Figure 4. The next step is to draw the points on the canvas and display the updated canvas and frame in their respective windows. If the user clicks the ‘CLEAR’ button, the code clears all the points from the arrays and the canvas.

### 3.4. Optical Character Recognition (OCR)

In our study, as shown in Figure 5, we delve into the intricate design of the Optical character recognition (OCR) system’s underlying structure [20]. This methodology unfolds in a carefully choreographed sequence, with each stage performing a distinct role: image acquisition, preprocessing, text recognition, and postprocessing. This primary stage aids as the cornerstone stage that includes converting the image into binary data. This transformative process is the pivotal mechanism that empowers the OCR system’s interpretive prowess. Optical character recognition (OCR) is important for accurately identifying the letters on elevator buttons, enabling service robots to navigate elevators effectively. Its integration with YOLOv5 improves recognition speed and accuracy [21]. Based on a study [22], OCR was of great benefit in its application to text along with optical mark recognition (OMR) for the digitization of music books [23].

In the subsequent preprocessing phase, a suite of techniques is harnessed, encompassing image alignment, noise reduction, and language identification. These endeavours collectively aim to refine the image by eliminating imperfections and enhancing its readiness for subsequent recognition. Moving forward, the third stage, text recognition, unfolds. Here, the OCR system diligently uses a combination of pattern matching and feature extraction algorithms. This concerted effort enables the system to discern and identify the intricate characters that compose the image. As we reach the culmination of this multi-stage endeavour, the post-processing stage comes to the forefront. The OCR system undertakes crucial tasks within this domain, including rectifications and format adjustments. The overarching goal is to bestow a mantle of accuracy and uniformity upon the recognized text, ensuring its fidelity. In the context of our study, a prominent role is assumed by the freely accessible online tool i2OCR. This tool is adept at extracting text from diverse sources, spanning images to scanned documents, encompassing materials such as books, faxes, contracts, invoices, mail, passports, and ID cards. An impressive spectrum of linguistic diversity, spanning over 100 languages, can be effectively deciphered.

### 3.5. Arabic Air-Writing Letter Recognition System Using Deep Convolutional Neural

We employed deep convolutional network models to improve performance and accuracy compared with classical models. This study specifically focused on combining convolutional neural network (CNN) models for Arabic letter image classification with machine learning algorithms, such as support vector machine (SVM). neural networks (NNs), random forest (RF), and K-nearest neighbour (KNN), to enhance the overall system performance and accuracy. Three different models were applied in this study, namely, VGG16, VGG19, and SqueezeNet, to assess their effectiveness in achieving the desired outcomes. By leveraging the strengths of both deep convolutional networks and machine learning models, this study provides a comprehensive approach to Arabic letter recognition that can have significant implications across various domains and applications.

#### 3.5.1. The VGGNet CNN Architecture

The graph VGG19 [24] presents a refined iteration of VGG16, incorporating a thoughtful modification. This enhancement introduces two supplementary convolutional layers, each boasting 512 filters and adhering to a kernel size of (3 × 3), all while maintaining a stride of 1. The activation function used in these layers remains consistent, embracing the ReLU framework [25]. A visual representation of both the VGG16 and VGG19 architectures is shown in Figure 6 and Figure 7.

Within the confines of this study, our focus zeroed in on the assessment of two distinct iterations of the VGGNet architecture: VGG16 and VGG19. The lineage of both VGG16 and VGG19 is traced back to their initial training on the extensive ImageNet dataset. This vast collection boasts over one million meticulously annotated images, spanned across an impressive gamut of 1000 classes. Their debut marked a watershed moment in image classification, as their performance soared to the zenith of the ImageNet classification task, setting a formidable benchmark at their inception. The graph shows VGG16 as a profound convolutional neural network meticulously tailored for image classification tasks. The inception of this network commences with the input layer, primed to accommodate an image of dimensions 224 × 224 × 3. Here, the width and height unfurl across 224 pixels, while the prism of colour channels remains RGB. The intricate lattice of this network is woven with an array of layers, spanning the domains of convolutional, max pooling, and fully connected counterparts. The inaugural duo is composed of convolutional layers, wherein each layer boasts a complement of 64 filters. These filters are cast over a kernel expanse of 3 × 3, harmoniously guided by a stride of 1. As the network forges ahead, the seventh and eighth strata don the attire of 512 filters, with the ninth and tenth layers echoing this design. In parallel, the VGG19 architecture emerges as a distinguished variant of VGG16, charting its course with an augmentation. This enhancement unfurls by incorporating two supplementary convolutional layers, each underpinned by a suite of 512 filters. These layers reverberate with a kernel realm of 3 × 3, coupled with a stride attuned to the pulse of 1. United by the ReLU activation function, these layers amplify the expressive capacity of the architecture. Illustrative renderings of the VGG16 and VGG19 in Figure 6 and Figure 7 epitomize a formidable embodiment of convolutional mastery.

#### 3.5.2. SqueezeNet Architecture

The SqueezeNet architecture, as shown in Figure 8, was introduced in [26]. It represents a convolutional neural network (CNN) tailored to excel in image classification tasks while being highly efficient. Its design incorporates multiple layers, including fully connected layers, convolutional and pooling layers, and fire modules, serving as compact and efficient building blocks to streamline the network’s size. At the input layer, raw images are fed into the network, and the convolutional layers play a crucial role in extracting distinctive features from these input images. To achieve this, a combination of 3 × 3 and 1 × 1 convolutional filters is utilized. Then, the pooling layers come into play, effectively reducing the size of the feature maps and exercising control over the risk of overfitting. This judicious combination of architectural elements empowers SqueezeNet to deliver exceptional performance while maintaining remarkable efficiency, making it an ideal choice for various image classification applications. Max pooling is used to downsize the feature maps, effectively reducing their size. The SqueezeNet architecture hinges on the pivotal fire modules, serving as its fundamental building blocks. These modules encompass a squeeze layer, which decreases the number of filters in the feature maps, and an expanded layer, which elevates the filter count using 1 × 1 and 3 × 3 convolutional layers. This ingenious design enables the network to maintain superior accuracy while being significantly smaller and faster than other CNNs. The ultimate classification decision is made by the fully connected layers, which comprise dense layers with a SoftMax activation function at the output layer. Figure 8 illustrates the architecture of SqueezeNet, demonstrating how its diverse layers and building blocks collaborate harmoniously to classify input images. Remarkably, the SqueezeNet architecture has been proven to excel in image classification tasks with remarkable efficiency. Its adept usage of fire modules, coupled with a combination of 3 × 3 and 1 × 1 convolutional filters, allows it to achieve high accuracy while maintaining a compact network size, rendering it an ideal choice for scenarios where computational resources are limited.

### 3.6. Hyperparameter Tuning

Hyperparameter tuning is pivotal in machine learning, especially when dealing with intricate models endowed with numerous parameters. Hyperparameters, being adjustable configurations, govern the learning process of a machine learning algorithm, standing apart from model parameters acquired during training. They directly influence the model’s performance and its aptitude to generalize effectively. Hyperparameter tuning entails methodically exploring and identifying the optimal combination of hyperparameters to optimize a model’s performance. The ultimate objective is to pinpoint the hyperparameter values that yield the highest accuracy, minimize error, or attain the best performance metric for a specific task. Among the various techniques for hyperparameter tuning, grid search and random search are two commonly used methodologies. A set of hyperparameters was initialized for each machine learning model, as shown in Table 1. The optimal hyperparameters for each model were obtained using the grid search optimization algorithm, which contributed to raising the accuracy and efficiency of the models by searching for the best hyperparameters.

#### 3.6.1. Grid Search

Grid search serves as a fundamental technique for optimizing hyperparameters in machine learning [27]. This method can train models using various hyperparameter combinations, subsequently evaluating their performance to discover the most promising configurations, as shown in Algorithm 1.
**Algorithm 1** Pseudo code for grid search**1.** **Function Grid Search():**2. Hyperparameter Grid Search = Define Hyperparameter Grid Search3. Best Hyperparameters = None4. Best Performance = Select5. for Hyperparameter in Hyperparameter Grid Search6. Model = Set Hyperparameters in Model7. Performance = Evaluate Model8. if Performance > Best Performance9.  Best Performance = Performance10.  Best Hyperparameters = Hyperparameters11. end if12. end for**13.** **END**

The process commences by selecting a specific set of hyperparameters for the model, which is then used in training and cross-validation. Grid search identifies configurations with the highest performance during validation by systematically exploring the hyperparameter space. These superior hyperparameter combinations are seamlessly integrated into the model, guaranteeing optimal performance. In our implementation, we opt for K = 5 in cross-validation, dividing the dataset into five subsets for thorough evaluation. The Algorithm 1 provides a detailed illustration of the grid search operation. This sophisticated approach proves to be a powerful tool in attaining finely tuned hyperparameters, thereby significantly boosting the performance and efficiency of our machine learning model. By methodically exploring diverse hyperparameter combinations, grid search allows us to pinpoint the best set of hyperparameters, leading to optimal model performance.

#### 3.6.2. Random Search

Based on the experiments conducted by [28], random search has been shown to outperform grid search in hyperparameter optimization. This method demonstrates exceptional efficiency in finding optimal models while minimizing computational time. Unlike grid search, random search explores broader areas by randomly sampling hyperparameter combinations. Each set of hyperparameters is then evaluated, allowing the method to discover promising configurations efficiently. The critical advantage of random search lies in its ability to explore hyperparameter spaces randomly, which often leads to the discovery of valuable configurations quickly. This streamlined approach presents a compelling alternative to grid search, showcasing its effectiveness in optimizing hyperparameters for machine learning models. Researchers and data scientists can benefit from adopting random search as a powerful tool to fine-tune their models and achieve enhanced performance with reduced computational overhead.

### 3.7. Supervised Machine Learning Models

This paper leverages machine learning (ML) methods to train and assess character image classification models. To achieve peak performance, meticulous optimization of the model’s parameters occurred during its developmental phase. Using 10-fold cross-validation, the training set, comprising 80% of the original dataset, was subjected to rigorous evaluation alongside the testing set of 20%. Optimal parameters yielding the highest accuracy were then incorporated into the final model. This study explored four renowned ML classification techniques, namely, support vector machines (SVMs), neural networks (NNs), random forests (RF), and K-nearest neighbours (KNNs). Each method received comprehensive scrutiny as part of this study’s endeavours.

Support Vector Machine (SVM): SVM is a widely favoured machine learning algorithm suitable for classification and regression tasks. The core concept behind SVM revolves around identifying the hyperplane that optimally separates the various classes within a dataset. The SVM architecture is relatively straightforward and comprises a few fundamental components. To begin, SVM is trained on a set of labelled data, which serves as the basis for determining the hyperplane that best segregates the different classes. In pursuit of superior performance, we used two optimization techniques: grid search and random search, which helped us find the most optimal hyperparameters. The kernel, an essential part of SVM [29], is crucial in mapping the data to higher dimensions, allowing for better class separation. We experimented with various kernel types, including linear, polynomial, and sigmoid kernels, evaluating their effectiveness using search-based optimization algorithms. Finally, the best kernel among these options was chosen to optimize the model’s performance [30].

Neural Networks (NNs): A neural network, also known as an artificial neural network (ANN), is a machine learning model widely used for classification tasks. In our work, this state-of-the-art model consists of three layers of nodes: input layers, hidden layers with 256 neurons, and output layers. Each node applies a transfer function to the weighted sum of nodes from the previous layer, along with a bias term. During the training process, the network weights and parameters are updated repeatedly using the available dataset. Given the specific context of our study, involving multiple ratings among 18 categories, we used the SoftMax activation function. The primary goal of this function is to convert the network outputs into predictive probabilities for each class, which enhances its ability to deal with multiclass classification scenarios. The Rectified Linear Unit (ReLU) was used as the activation function, and the optimizer Adaptive Moment Estimation (Adam) was used, which is responsible for adjusting the learning rate for each parameter during training [31].

Random Forest (RF): Random forest (RF) is an important machine learning classifier known for its effectiveness and versatility, which belongs to group learning methods. It constructs multiple decision trees by selecting random samples from a dataset, and the accuracy is increased with the random construction of trees. Based on our study of the multiple classifications between letters, the voting method was used, and the result was obtained based on the highest number of votes [32].

K-Nearest Neighbour (KNN): KNN, or K-nearest neighbour, is a versatile learning algorithm for classification and regression tasks [33]. Its foundation lies in assessing the similarity between data points, and it operates by determining the value of K, a positive number representing the number of nearest neighbours to consider when calculating distances. The algorithm identifies the closest neighbours and assigns the category based on their votes by iterating between K and new data points. Choosing the appropriate value of K is crucial in KNN. A small K value may result in overfitting, where the model becomes too sensitive to noise and specific data points, leading to reduced generalization. On the other hand, a large K value can cause underfitting, where the model oversimplifies and fails to capture intricate patterns in the data. In our study, we determine the class of each letter based on the majority vote of its nearest neighbours, considering their proximity. The class with the highest vote becomes the predicted class for the letter under consideration.

### 3.8. Evaluation of Models

When assessing the performance of both deep learning (DL) and traditional machine learning (ML) models for classification, a range of metrics comes into play, and accuracy holds a pivotal position among them. The confusion matrix emerges as a critical player, furnishing vital details about actual and predicted labels, thereby facilitating an in-depth analysis of the model’s efficacy. To comprehensively evaluate the model’s performance, the confusion matrix yields invaluable insights by breaking down the counts of True Negatives (TNs), True Positives (TPs), False Negatives (FNs), and False Positives (FPs). These metrics are critical in measuring the model’s accuracy of distinctions across various classes. The accuracy is computed as depicted in Equation (1):(1)$$\;\mathrm{Accuracy}\;=\frac{TP+TN}{TP+TN+FP+FN}$$

## 4. Result and Discussion

In this section, we perform several setups and evaluations such as performance evaluation, accuracy mean scores, writing testing samples, overall model validation, and comparison to others. The upcoming subsections detail the experimental results.

### 4.1. Performance of Classification Algorithms

Table 2 outlines the performance outcomes achieved using various classifiers on both CNN models (VGG19, VGG16, and SqueezeNet) and ML models (SVM, NN, RF, and KNN) with three distinct optimization techniques: grid search, random search, and default parameters. The algorithms underwent thorough evaluation under different optimization approaches: grid search, random search, and default parameters. A closer examination of the performance disparities across diverse optimization strategies reveals noteworthy fluctuations in accuracy across different models and classifiers, as demonstrated in Figure 9. The comparison of accuracy among ML classifiers yields intriguing insights. Notably, grid search produces an accuracy of 0.888 for the NN classifier when applied to the VGG16 model, closely followed by SVM, with an accuracy of 0.855. However, the accuracy decreases to 0.843 when default parameters are used.

For the NN classifier, all optimization techniques yielded remarkable results, consistently achieving accuracy above 0.847 in VGG19. Both SVM and NN classifiers exhibited outstanding precision when applied to the VGG16 model. Specifically, the NN classifier attained the highest accuracy under grid search, registering 0.888, followed by SVM at 0.855 and default parameters at 0.843. The comparison of performance across various optimization methods provides valuable insights into the efficacy of different approaches in fine-tuning the models for improved accuracy and precision. The NN classifier exhibited commendable performance with accuracy values of 0.888, 0.847, and 0.843 under grid search, random search, and default parameters, respectively. In the SqueezeNet model, the NN classifier achieved an accuracy of 0.813 using default parameters, 0.823 with random searches, and 0.825. The NN classifier consistently showcased outstanding results across all optimization techniques, maintaining accuracy surpassing 0.888. The SVM classifier consistently outperformed CNN models, particularly VGG19 and VGG16. The neural network classification consistently outshined other classifiers across every model and optimization technique for the AHAWP dataset. The RF and KNN classifiers generally demonstrated lower accuracy than the SVM and NN classifiers.

For Arabic handwritten recognition on the AHAWP dataset, the NN classifier continuously demonstrated the highest accuracy rates, reaffirming its potential in this domain. Results from the SVM classifier were also promising, mainly when used with the VGG16 model. However, future research should explore areas for improvement, as the RF and KNN classifiers exhibited lower accuracy results. The optimization approach significantly impacted accuracy, with grid search and random search consistently outperforming default parameters. The t-tests conducted in this investigation compared the mean accuracy scores of several pairs of models, providing insights into their statistical significance. The calculated *p*-values indicated the likelihood of observed differences in mean accuracy being genuine, with a lower significance level indicating more substantial evidence against the null hypothesis. When comparing the SVM classifier with other classifiers (NN, RF, and KNN) using the VGG16 CNN model, the *p*-value of 0.861 exceeded the conventional significance threshold of 0.05.

The *p*-values, as shown in Table 3 for the comparisons between SVM and RF and SVM and KNN, are less than 0.05, which shows that the accuracy achieved with the SVM classifier differs significantly. The *p*-values from the t-tests give us essential information about the statistical significance of the variations in mean accuracy between the models.

These results add to our comprehension of the related results in Figure 10 and allow us to conclude that the SVM classifier performs similarly to the NN classifier. However, when utilizing the VGG16 CNN model, it dramatically surpasses the RF and KNN classifiers when it comes to accuracy. The Tukey’s honestly significant difference (HSD) test results yield significant insights regarding the pairwise differences in mean accuracy among the examined models, as shown in Table 4. Comparing the KNN model to the NN model, we observe a mean difference of 0.1173. The associated *p*-value is 0.0006, which falls below the significant threshold of 0.05. Consequently, we can conclude a statistically significant difference in mean accuracy between the KNN and NN models. Specifically, the KNN model exhibits a higher mean accuracy than the NN model. However, when comparing the KNN and RF models, we discover a mean difference of 0.09 with a *p*-value of 0.009519. As the *p*-value goes above 0.05, we lack enough evidence to support a significant difference in mean accuracy among the KNN and RF models.

Additionally, upon closer examination of the KNN and SVM models, a mean difference of 0.115 with a *p*-value of 0.0007 becomes evident. Given that the *p*-value is below the significance threshold of 0.05, this indicates a statistically significant divergence in mean accuracy between the KNN and SVM models. Specifically, the KNN model showcases a notably higher mean accuracy than the SVM model.

Turning our attention to the contrast between the NN and RF models, a mean difference of −0.1083, denoted by the negative sign, emerges. This suggests that the NN model displays a slightly lower mean accuracy than the RF model. The corresponding *p*-value of 0.0011 further emphasizes a significant dissimilarity in mean accuracy between these two models, with the RF model demonstrating superior performance.

Conversely, the mean difference between the NN and SVM models is −0.0023, a value proximate to zero. The associated *p*-value of 0.999 implies that the mean accuracy of the NN and SVM models is essentially indistinguishable. As such, any observed fluctuations in accuracy between these models are more likely attributable to chance than a substantive performance variance.

Shifting our focus to the mean-variance analysis between the RF and SVM models, a mean-variance of 0.106 is identified, accompanied by a *p*-value of 0.0013. This *p*-value corroborates a significant discrepancy in mean accuracy. The outcomes of Tukey’s HSD test illuminate the nuances in the dissimilarities among the evaluated models in terms of mean accuracy. Figure 10 graphically elucidates the superiority of the NN model in mean accuracy, surpassing the other models. However, a discernible discrepancy in mean accuracy between the NN and SVM models is not discerned.

When juxtaposed with the alternative models, the NN model, the top performer, demonstrates enhanced accuracy in accurately classifying air-written letters, as shown in Figure 11. This suggests that the model can adeptly detect and assign appropriate labels to the letters. Figure 12, Figure 13, Figure 14 and Figure 15 illustrate the confusion matrix for the ML models, identifying the ML models as the most proficient contenders. The confusion matrix provides insightful revelations into the model’s predictions, allowing us to gauge the precision of classifying air-written letters. Using the confusion matrix, we can meticulously assess the model’s classification accuracy, identify any errors or zones of uncertainty, and decode the matrix where anticipated letter labels align with the columns while actual letter labels align with the rows.

### 4.2. Samples of Letters Air Writing

Our experiments aimed to develop a model combining machine learning (ML) and optical character recognition (OCR) methods to identify Arabic letters written in the air accurately. For the experimental setup, air writing was recorded using a laptop equipped with a Core i5 10th generation camera. We used the Python 3.12.0 programming language and PyCharm development environment. We used an NVIDIA GeForce GT graphics card to enhance performance. Figure 16 showcases a sample of the air-written lettering captured during the experiment.

### 4.3. Model Validation

Validation of the model involved an elaborate process designed explicitly for air-written Arabic letters, as shown in Figure 17. The procedure commenced with capturing the letter as an image, which was then saved for further processing. The image underwent a series of preprocessing steps to enhance clarity and eliminate unwanted artifacts. Among these steps were resizing the image and using smoothing techniques to refine the letter’s edges and visibility. Once the preprocessing phase was complete, the letter image was fed into a machine learning algorithm meticulously trained on a dataset containing images of Arabic letters. This algorithm was purpose-built to leverage the knowledge gained from the training dataset’s patterns and features, enabling it to effectively predict and classify the input letter. During the validation phase, the model used a trained algorithm to make precise predictions for an air-written letter. The model skilfully identified its corresponding Arabic character by comparing the input letter with the patterns and characteristics it internalized during training. This prediction process proved invaluable in accurately recognizing and classifying the air-written letters, showcasing the robustness and efficacy of our developed approach.

The validation results were evaluated based on the accuracy of the model’s predictions, as shown in Figure 14. The accuracy represented the percentage of correctly identified letters from the total validation set. This metric served as an indicator of the model’s performance and ability to recognize air-written Arabic letters accurately. The validation process played a crucial role in assessing the reliability and effectiveness of the developed model. It allowed us to verify the model’s ability to recognize individual Arabic characters based on air writing input accurately. By comparing the model’s predictions with the ground truth labels, we determined the level of accuracy achieved and identified any areas for improvement. Our model’s performance in recognizing air-written Arabic letters was assessed using predicted letter pairs. The true/false category was verified based on whether the model correctly detected the letter or not. The results in Table 5 indicate that the model achieved a high level of accuracy in some cases, correctly predicting the letters Beh (ب), Ain (ع), Heh (ه), Jeem (ج), Kaf (ك), Meem (م), Noon (ن), Raa (ر), Sad (ص), Seen (س), and Tah (ت). However, there were instances where the model made incorrect predictions, such as misclassifying Dal (د) as Ain (ع), Feh (ف) as Qaf (ق), Lam (ل) as Dal (د), and Waw (و) as Heh (ه).

### 4.4. A Comparison Study

Table 6 compares the approaches and findings from earlier studies in air-writing recognition. In our research, we suggested a model that combines machine learning (ML) and optical character recognition (OCR) methods to recognize Arabic air writing specifically. Even if our model’s accuracy is lower than some of the earlier research included in Table 6, it is crucial to consider the particulars of the Arabic script and its difficulties. Our study stands out as the first to concentrate on recognizing Arabic air writing. 

This demonstrates its substantial impact on the industry. However, an 88% accuracy rate may seem lower compared with research focusing on the English language or other languages. The accuracy levels presented in the table should be interpreted with caution, as direct comparisons may not be appropriate due to variations in datasets, techniques, and language-specific characteristics. Each study examines a different language and writing system, necessitating unique methodologies and considerations. Despite these differences, our model’s effectiveness in identifying Arabic air-written letters is evident through its impressive performance, demonstrating its value within the context of Arabic script. Also, some related studies including Refs. [8,10,15] classified symbols, gestures, and diacritics using CNN architecture and ML models. Our research addresses a significant gap in air-writing identification by focusing on Arabic script, which paves the way for future advancements and practical applications. This highlights the importance of devising tailored strategies that address the specific challenges posed by various writing systems and languages. Moreover, our study contributes to the knowledge base of Arabic air writing and establishes itself as the first investigation into Arabic air-writing recognition. While previous studies have primarily focused on English or other languages, we recognized the importance of understanding Arabic script’s unique complexities and characters. By exploring this specific context, our study enriches the knowledge of air writing recognition techniques for the Arabic language. Our model effectively combines machine learning (ML) and optical character recognition (OCR) methods to accurately recognize Arabic air-written letters, offering a comprehensive solution that leverages the strengths of both approaches. As a pioneer in Arabic air writing recognition, our research lays a solid foundation for future investigations. This opens new avenues for researchers to explore additional methods and techniques to enhance recognition accuracy in Arabic air writing. As we move forward, these findings will undoubtedly spur further advancements and innovations in air-writing recognition for Arabic and other languages.

## 5. Conclusions and Future Work

This paper presented a new method for recognizing air-written Arabic letters. We used machine learning (ML) models and CNNs with two optimization methods: grid search and random search. The results showed that the NN method, along with the features extracted from VGG16 with the grid search optimization algorithm, achieved high accuracy compared with the default parameters of the model. Further, NN had low error rates in recognizing Arabic letters. The NN model’s overall accuracy was 0.88 using the grid search optimization algorithm and 0.847 using random search, 0.843. Our study also explored using optical character recognition as an alternative method for identifying air-written letters, which shows similar results to machine learning models. Notably, our model performed exceptionally well on the AHAWP dataset and showed improved results compared with other models. This research contributes to expanding the scope of knowledge in handwriting recognition in Arabic. The approach offers potential applications in education and helping individuals with physical communication disabilities. One of this study’s limitations was the need for more data containing handwritten Arabic letters. In addition, the dataset that was used in this study only includes some of the letters of the Arabic language. Future studies should consider studying hybrid machine learning and deep learning approaches. We also seek to collect more data on all Arabic letters in the future and make the approach accessible to other researchers.

## Figures and Tables

**Figure 1 sensors-23-09475-f001:**
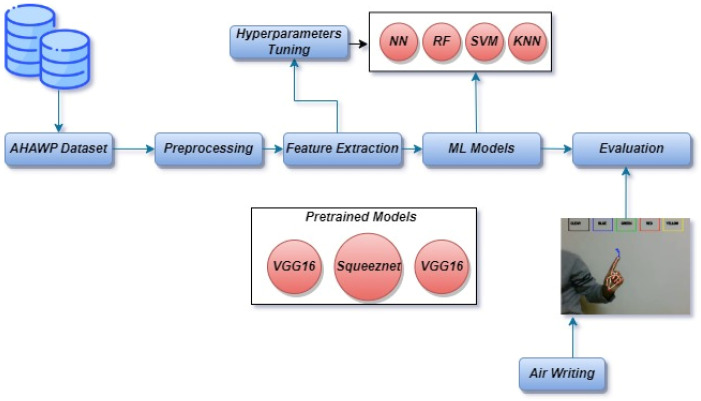
Architecture of the proposed approach.

**Figure 2 sensors-23-09475-f002:**
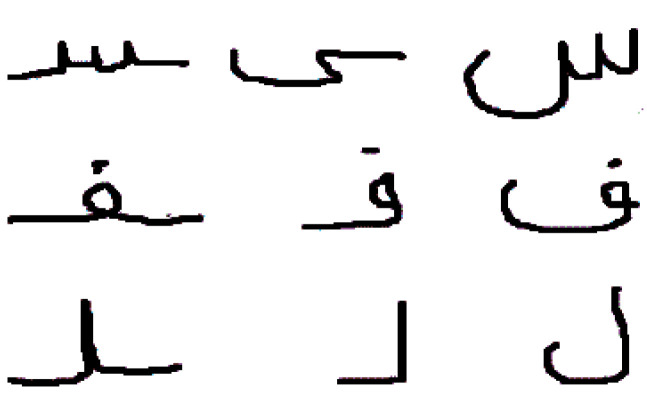
Sample from the dataset.

**Figure 3 sensors-23-09475-f003:**
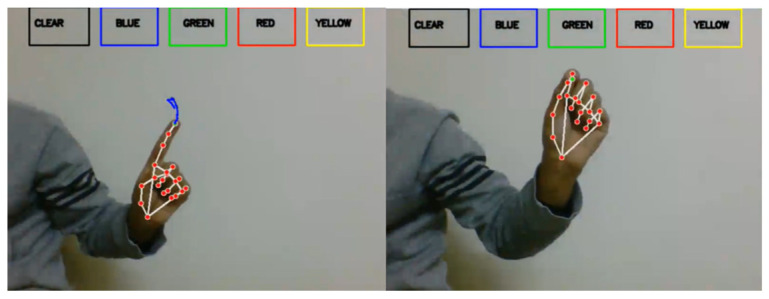
This example shows a type of writing starting on the left and stopping on the right.

**Figure 4 sensors-23-09475-f004:**
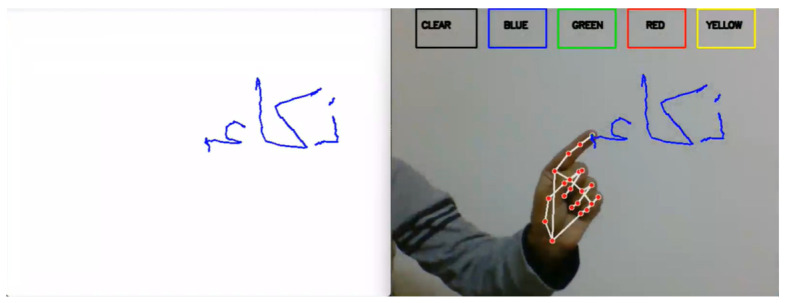
Result of writing.

**Figure 5 sensors-23-09475-f005:**
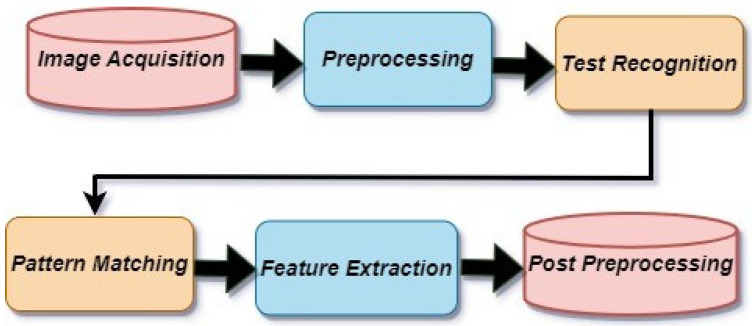
Architecture of the OCR.

**Figure 6 sensors-23-09475-f006:**
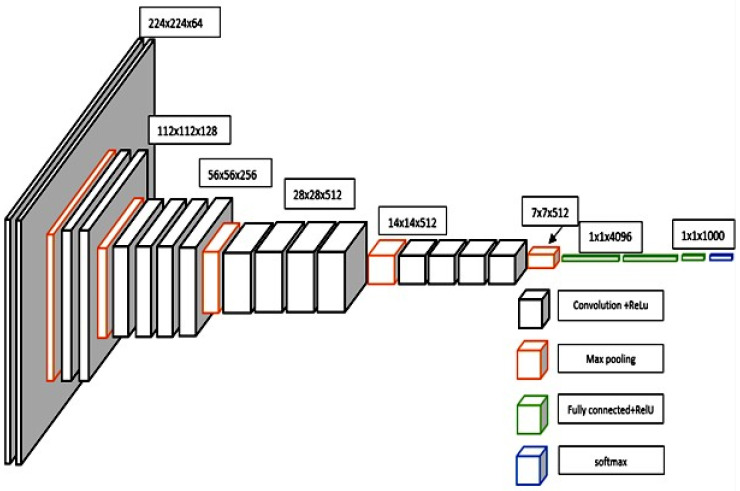
Architecture of VGG19.

**Figure 7 sensors-23-09475-f007:**
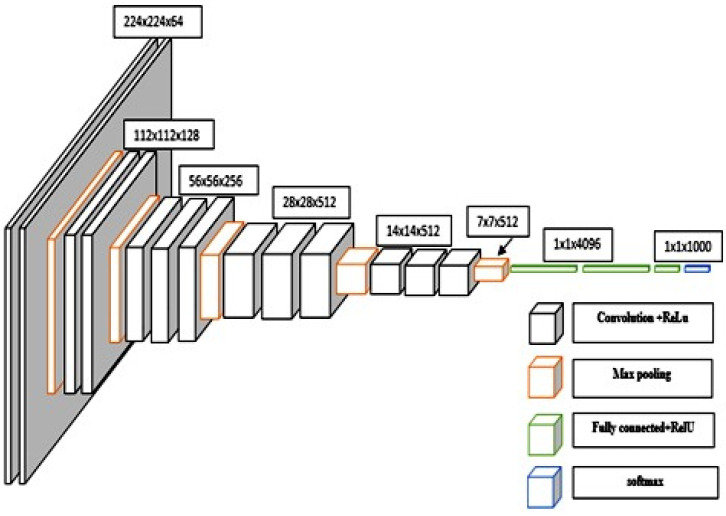
Architecture of VGG16.

**Figure 8 sensors-23-09475-f008:**
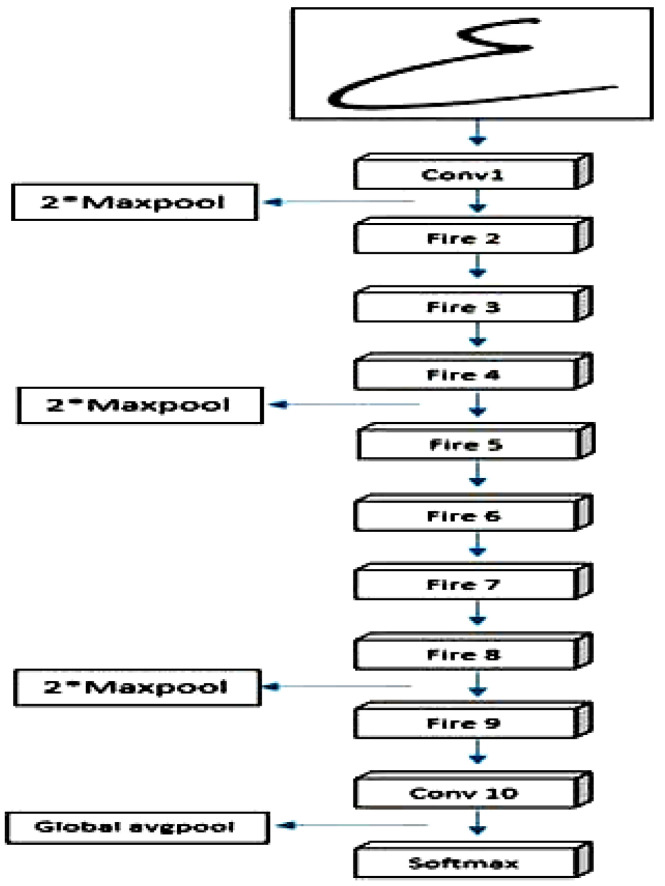
Architecture of SqueezeNet.

**Figure 9 sensors-23-09475-f009:**
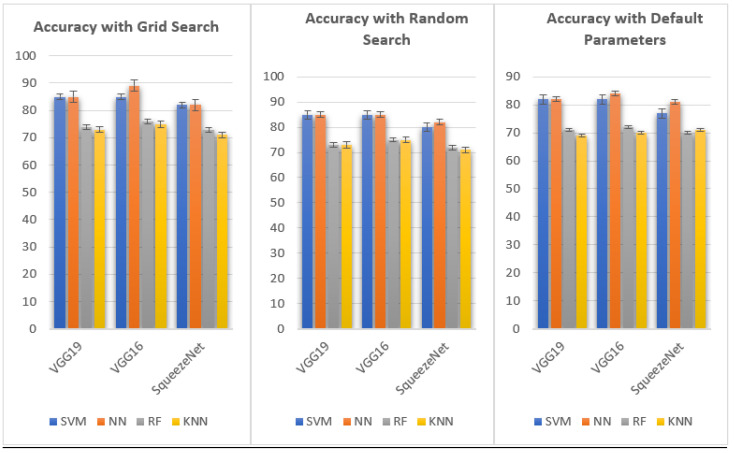
Comparison of the accuracy of the ML classifier performed.

**Figure 10 sensors-23-09475-f010:**
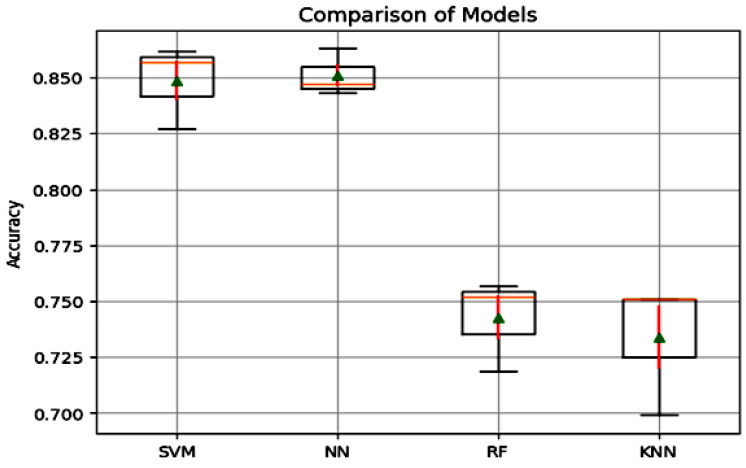
Comparison box plot of ML models.

**Figure 11 sensors-23-09475-f011:**
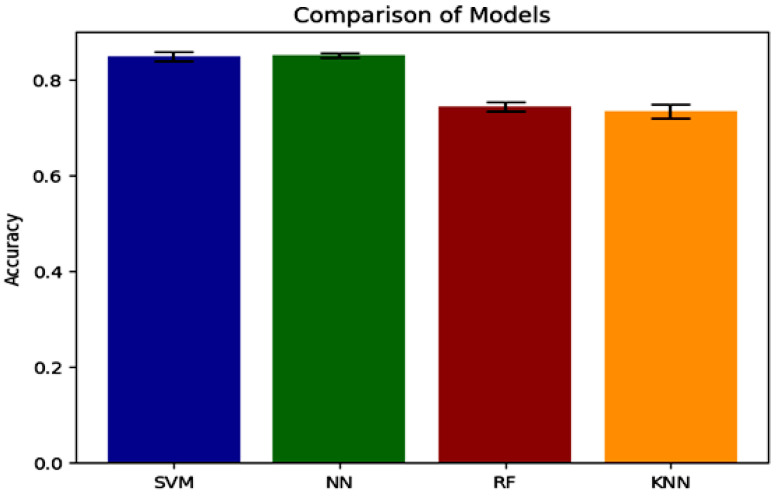
Accuracy comparison between models with VGG16 using grid search.

**Figure 12 sensors-23-09475-f012:**
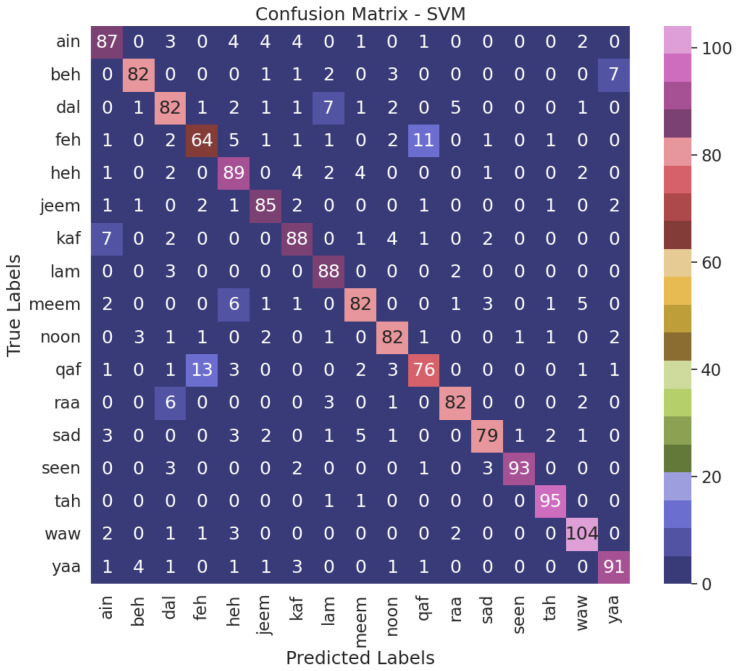
Confusion matrix for the SVM model with VGG16 using grid search.

**Figure 13 sensors-23-09475-f013:**
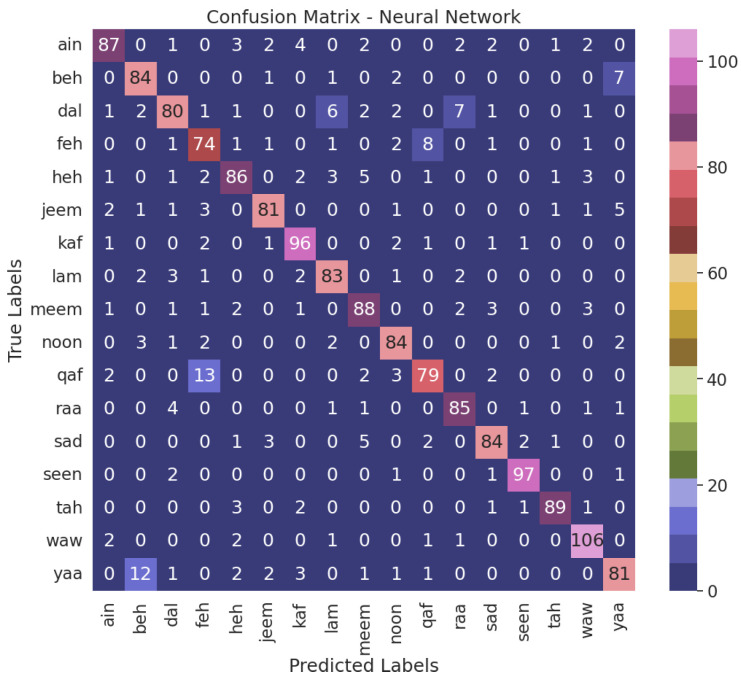
Confusion matrix for the NN model with VGG16 using grid search.

**Figure 14 sensors-23-09475-f014:**
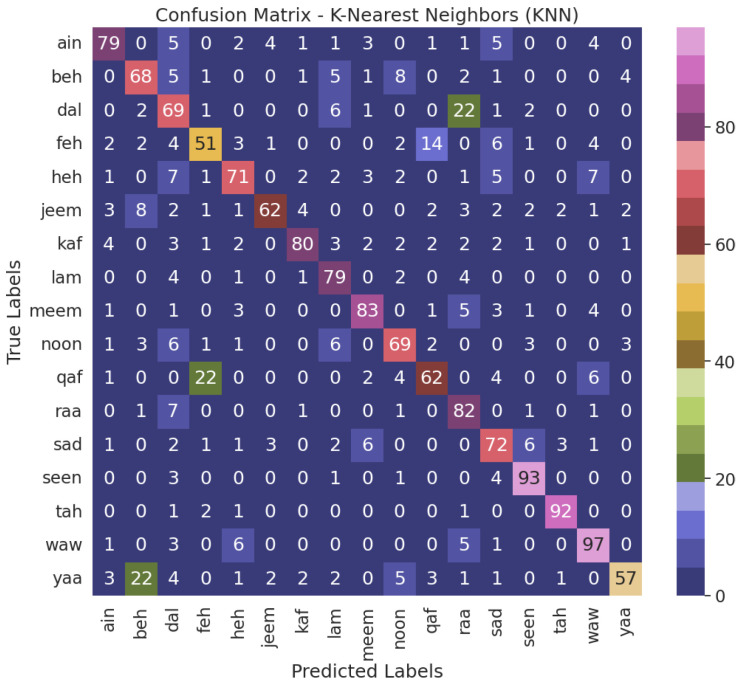
Confusion matrix for the KNN model with VGG16 using grid search.

**Figure 15 sensors-23-09475-f015:**
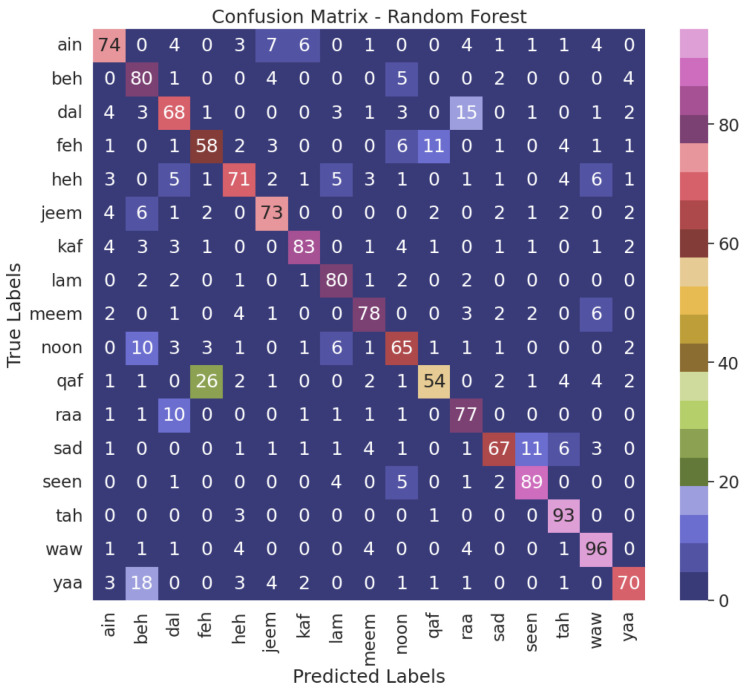
Confusion matrix for the RF model with VGG16 using grid search.

**Figure 16 sensors-23-09475-f016:**
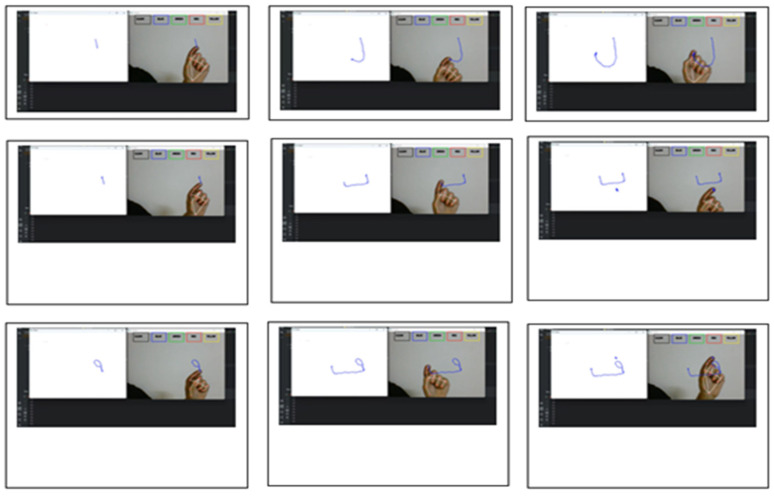
Sample of lettering writing.

**Figure 17 sensors-23-09475-f017:**
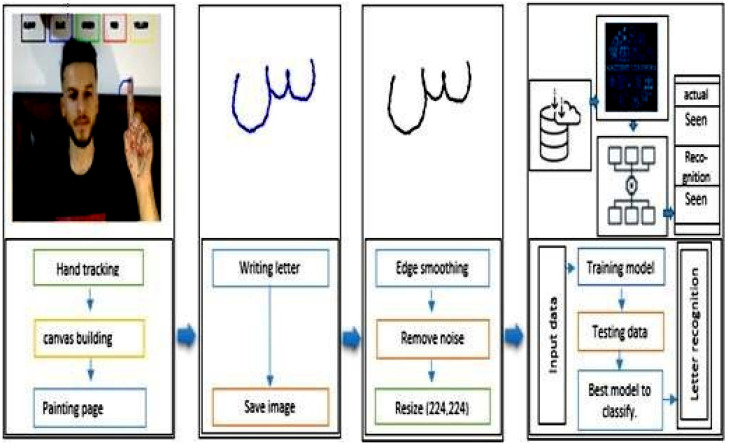
Validation process for air-written Arabic letters.

**Table 1 sensors-23-09475-t001:** Hyperparameters for ML models.

ML Model	Hyperparameters Search	Best Hyperparameters (Grid Search)
NN	hidden_layer_sizes: [(50,), (100,), (200,)]; activation: [relu, ‘tanh’]; solver: [adam, sgd]; alpha: [0.0001, 0.001, 0.01]; learning rate: [0.001, 0.01, 0.1].	hidden_layer_sizes: 200; activation: relu; solver: adam; alpha: 0.001; learning rate: 0.01.
SVM	C: [0.1, 1, 10]; kernel: [linear, rbf]; gamma: [0.0001, 0.001, 0.01].	C: 10; kernel: rbf; gamma: 0.01.
RF	n_estimators: [50, 100, 200]; max_depth: [None, 10, 20]; min_samples_split: [2, 5, 10]; min_samples_leaf: [1, 2, 4].	n_estimators: 200; max_depth: 20; min_samples_split: 5; min_samples_leaf: 1.
KNN	n_neighbors: [3, 5, 7]; weights: [uniform, distance], p: [1, 2].	n_neighbors: 5; weights: distance, p: 1.

**Table 2 sensors-23-09475-t002:** Performance of classification of algorithms.

CNN Model	ML Model	Grid Search	Random Search	Default Parameters
VGG19	SVM	0.855	0.853	0.816
	NN	0.847	0.851	0.825
	RF	0.744	0.735	0.706
	KNN	0.727	0.727	0.692
VGG16	SVM	0.855	0.853	0.816
	NN	0.888	0.847	0.843
	RF	0.757	0.752	0.719
	KNN	0.751	0.751	0.699
SqueezeNet	SVM	0.819	0.799	0.770
	NN	0.825	0.823	0.813
	RF	0.729	0.724	0.695
	KNN	0.712	0.712	0.632

**Table 3 sensors-23-09475-t003:** Comparison of ML *t*-tests and *p*-values.

*t*-Test	*p*-Value
SVM vs. NN	0.86123788
SVM vs. RF	0.00280216
SVM vs. KNN	0.00495305

**Table 4 sensors-23-09475-t004:** The Tukey’s honestly significant difference (HSD) test results.

Reject	Upper	Lower	p-adj	Mean Diff	Group2	Group1
True	0.1728	0.0619	0.0006	0.1173	NN	KNN
True	0.0644	−0.0464	0.009519	0.09	RF	KNN
True	0.1704	0.0596	0.0007	0.115	SVM	KNN
True	−0.0529	−0.1638	0.0011	−0.1083	RF	NN
False	0.531	−0.0578	0.999	−0.0023	SVM	NN
True	0.1614	0.0506	0.0013	0.106	SVM	RF

**Table 5 sensors-23-09475-t005:** Actual and Prediction Examples.

Actual	Prediction	True/False
Beh (ب)	Beh (ب)	T
Dal (د)	Ain (ع)	F
Ain (ع)	Ain (ع)	T
Feh (ف)	Qaf (ق)	F
Heh (ه)	Heh (ه)	T
Jeem (ج)	Jeem (ج)	T
Kaf (ك)	Kaf (ك)	T
Lam (ل)	Da l(د)	F
Meem (م)	Meem (م)	T
Noon (ن)	Noon (ن)	T
Qaf (ق)	Feh (ف)	F
Raa (ر)	Raa (ر)	T
Sad (ص)	Sad (ص)	T
Seen (س)	Seen (س)	T
Tah (ت)	Tah (ت)	T
Waw (و)	Heh (ه)	F
Yaa (ي)	Beh (ب)	F

**Table 6 sensors-23-09475-t006:** A comparison of the proposed model with preview work.

Paper	Languages	Method	Result
[1]	Air writing English	2D-CNN	Accuracy: 91.24%
[2]	Air writing English	MS-CNN	Accuracy: 95%
[3]	Air writing English	LSTM	Accuracy: 99.32%
[4]	English	Faster RCNN	Accuracy: 94%
[5]	Air writing Korean and English	3D ResNet	Character error rate (CER): Korean: 33.16%, English: 29.24%
[8]	Numeric symbols (zero to nine)	1D-CNN and 2D-CNN	Accuracy: 99%
[10]	Diacritics and Ottoman font in Arabic script	CNN and RNN	Accuracy: 98%
[12]	Air writing English	Faster RCNN	Mean accuracy: 96.11%
[15]	gestures	SVM, kNN, Naïve Bayes, ANN, and ELM	Accuracy of SVM: 96.95
[34]	Air writing English		Error rate: 0.8%
[35]	Air writing Hindi	PointNet	Recognition rate: >97%
Our Model	Air writing Arabic	Hybrid Model VGG16 + NN	Accuracy: 88%

## Data Availability

Data are contained within the article.

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
