# Peer review of "Recognition of Arabic Air-Written Letters: Machine Learning, Convolutional Neural Networks, and Optical Character Recognition (OCR) Techniques"

_sensors, 2023, doi:10.3390/s23239475_

Round 1

Reviewer 1 Report

Comments and Suggestions for Authors

Paper deals with important task. It has a scientific novelty. It has a logical structure. The paper is technically sound. The proposed approach is logical. The description of the results should be improved.

Suggestions: 

1.    The authors use PCA as a data dimensionality reduction method. However, an extended description of the results of this method is not provided. What is the value of the explained variance, what gain did this approach give in training speed and possibly accuracy?

2.    The article uses a combination of CNN and SVM, NN, RF and KNN. The best result is obtained for the combination of CNN and NN. However, the structure of the NN, its parameters, and the methodology for choosing a specific NN architecture are not described.

3.    Methods such as Grid Search and Random Search were used to select hyperparameters. However, the found optimal values of hyperparameters are not indicated. Also, the range in which the search was conducted is not specified.

4.    The number of samples included in the confusion matrix in Fig. 10 are significantly different. The authors do not explain clearly enough why this is so. How are test datasets formed?

5.     It would be appropriate to expand the overview of the application of the combining OCR methods for character recognition problems. For example, pay attention to publications DOI: 10.32604/cmc.2023.033327 DOI: 10.1007/978-3-031-06555-2_37 and DOI: 10.1007/978-3-031-24475-9_56

6.    Conclusion section should be extended using: 

- numerical results obtained in the paper;

- limitations of the proposed approach.

Author Response

I want to express my gratitude for your thorough review of our manuscript and for providing us with valuable insights. Your feedback has been instrumental in improving our work, and we've implemented several significant changes based on your suggestions.

Reviewer 2 Report

Comments and Suggestions for Authors

The paper addresses the challenging problem of recognizing Arabic air-written letters, a topic of growing interest. It introduces a model that combines machine learning (ML) techniques with optical character recognition (OCR) methods, utilizing various ML algorithms and deep convolutional neural networks (CNNs) for feature extraction. The study employs the AHAWP dataset and demonstrates promising results in terms of accuracy.

The paper, titled "Recognition of Arabic Air-Written Letters," addresses an intriguing and relevant research problem - recognizing Arabic letters written in the air. The topic is undoubtedly of interest, given its potential applications in gesture recognition and accessibility technologies. However, while the paper touches on several important aspects, it falls short in several critical areas:

The paper suffers from a lack of clarity in its structure and presentation. The abstract, in particular, needs refinement. The objectives and contributions of the study should be explicitly stated, providing readers with a clear roadmap of what to expect. The introduction should also provide more context on the significance of this research problem.

The reviewers' concern about the quality of images and diagrams is valid. High-quality visuals are essential in a technical paper, especially when dealing with complex algorithms and methodologies. Improving the clarity of these visuals would significantly enhance the paper's overall readability.

The paper briefly mentions the use of ML algorithms and deep CNNs but lacks sufficient detail in explaining the methods, parameters, and hyperparameters employed. A more comprehensive explanation of the model architecture, training process, and feature extraction would greatly benefit readers who may want to replicate or build upon this work.

While the paper mentions the use of the AHAWP dataset and reports an accuracy of 88.8%, it lacks a detailed description of the experimental setup. This includes the preprocessing techniques applied, data augmentation (if any), and a clear delineation of the training and testing procedures.

The paper would benefit from a more comprehensive literature review that not only provides context but also highlights the gaps and limitations of existing approaches in recognizing air-written Arabic letters. This would strengthen the paper's contribution to the field.

The paper should address the broader impact of its findings and potential applications beyond the immediate context. How could this technology be used in real-world scenarios? What are the implications for accessibility or other fields?

The paper contains numerous grammatical errors and awkward phrasing. Careful proofreading and editing are essential to improve the overall quality of the writing.

Comments on the Quality of English Language

There are notable weaknesses that need attention. Firstly, the section includes several sentences that are quite lengthy and contain multiple clauses, making them challenging to follow. Breaking these sentences into shorter, more digestible units would greatly improve readability.

Additionally, there are instances of grammatical issues, including sentence fragments and awkward phrasing. These issues need correction for the text to read more smoothly and coherently.

Moreover, there is occasional redundancy in the text, with ideas being repeated or redundant phrases being used. Streamlining these portions would enhance clarity.

The section also features complex sentence structures that, while dealing with intricate concepts, can be simplified for better comprehension.

Transitions between sentences and ideas could be improved to create a smoother flow in the narrative. This would make it easier for readers to follow the logical progression of the introduction.

Furthermore, the introduction is somewhat lengthy and could benefit from better organization into shorter paragraphs, each focusing on a specific aspect of the research problem or its significance.

Lastly, while citations are essential, they should be integrated seamlessly into the text to avoid disrupting the flow of the narrative. Consider incorporating them more smoothly.

Author Response

(The authors gave the same response as above.)

Reviewer 3 Report

Comments and Suggestions for Authors

The content of the article is in accordance with the scientific field of Sensors magazine.
The topic raised by the authors is current.
The authors address the interesting topic of recognizing aArabic Air-Written Letters.

The authors proposed a hybrid model: feature extraction with deep learning models and machine learning and optical character recognition methods, and applied grid and random search optimization algorithms.

The contribution has an original and research character. It is appropriately structured, supplemented by appropriate images.
Results and conclusions are clearly described.

The paper has already been reviewed and modifications have been incorporated into it.

I recommend publishing the paper.

Author Response

I want to express our gratitude for your thoughtful review of our manuscript. Your feedback is highly valuable to us.

Reviewer 4 Report

Comments and Suggestions for Authors

The paper is interesting and novel, however I have some amendments.

- Line 88. "Impressively, this model achieved an impressive accuracy" is redundant.

- Line 109. "%" missing

-Line 162. "This research’s main contribution is to 162 practically contribute" is redundant.

- Fig. 1. Please enhance the quality of the figure and explain its content by text.

- Line 198. Did you mean "prepROCessing"?

- Figs 3 and 4. Please enhance the quality of the figure.

- Line 302. Please cite the datasets reference.

- Figs 6,7,8. Please enhance the quality of the figure.

- Table 3. "NN" is not visible.

- Fig. 9.  Please enhance the quality of the figure, the accuracies are not easily readable.

- Fig 10 and 11. Please enhance the quality of the figure.

- Fig 12. Consider splitting in the figure in 4 separate ones.

- Fig 13 and 14. Please enhance the quality of the figure.

- The Authors should consider adding a comparison to "non-aerial" Arabic language classifiers.

Comments on the Quality of English Language

I found several typos.

Author Response

I want to express our gratitude for your thoughtful review of our manuscript. Your feedback is highly valuable to us, and we've taken significant steps to address your comments. Let me walk you through the changes we've made:

Round 2

Reviewer 1 Report

Comments and Suggestions for Authors

The authors take into account the recommendations.

Author Response

(The authors gave the same response as above.)

Reviewer 4 Report

Comments and Suggestions for Authors

I would like to thank the Authors to their reply to my concenerns. However, I still find that many of the Figures have a very low resolution and their aspect ratio may not be correct. Please update all of them.

Author Response

Thank tou very much for your recommendation
